# Association between Glycosylated Hemoglobin Levels and Vaccine Preventable Diseases: A Systematic Review

**DOI:** 10.3390/diseases12080187

**Published:** 2024-08-17

**Authors:** Elda De Vita, Federica Limongi, Nicola Veronese, Francesco Di Gennaro, Annalisa Saracino, Stefania Maggi

**Affiliations:** 1Clinic of Infectious Diseases, Department of Precision and Regenerative Medicine and Ionian Area (DiMePRe-J), University of Bari Aldo Moro, 70124 Bari, Italy; elda.devita@uniba.it (E.D.V.); francesco.digennaro1@uniba.it (F.D.G.); annalisa.saracino@uniba.it (A.S.); 2Aging Branch, CNR Institute of Neuroscience, 35127 Padua, Italy; federica.limongi@in.cnr.it (F.L.); stefania.maggi@in.cnr.it (S.M.); 3Department of Internal Medicine and Geriatrics, University of Palermo, Via del Vespro 141, 90127 Palermo, Italy

**Keywords:** diabetes, glycosylated hemoglobin, vaccine-preventable diseases, herpes zoster, COVID-19

## Abstract

People with diabetes are at higher risk of serious complications from many vaccine-preventable diseases (VPDs). Some studies have highlighted the potential impact of glycosylated hemoglobin levels (HbA1c), but no systematic review has synthesized these findings. Of the 823 identified studies, 3 were included, for a total of 705,349 participants. Regarding the incidence of herpes zoster (HZ), one study found that higher HbA1c levels at the baseline (>10.3%) were associated with a significantly higher risk of HZ of 44%, compared to those with a good HbA1c control (6.7%). On the contrary, the second one reported that when compared to the reference group (HbA1c of 5.0–6.4%), participants with a HbA1c less than 5.0% were at higher risk of HZ of 63%, whilst participants with a HBA1c more than 9.5% had a similar risk. Finally, the third study observed that diabetes, defined using a value of HbA1c more than 7.5%, was associated with an increased risk of mortality in men with COVID-19. In conclusion, both high and low HBA1c levels appear to be associated with a higher risk of HZ. Regarding COVID-19, a value of HbA1c more than 7.5% was associated with a higher risk of death in COVID-19, but only in men.

## 1. Introduction

According to the Global Burden of Disease, there were 529 million people with diabetes in the world in 2021. This figure is projected to reach approximately 1.31 billion by 2050 [1]. Diabetes, which represents one of the principal causes of morbidity and death in the world [1], is associated with a higher risk of developing various types of infections [2], as well as of hospitalization and infection-related mortality [3]. A review of 13 population-based studies and 1 randomized controlled study reported a strong association between higher glycosylated hemoglobin (HbA1c) levels and infectious diseases (IDs) in both type 1 and type 2 diabetes (T1DM and T2DM) [4]. Furthermore, a high HbA_1c_ level (>8.5%) has been identified as a predictor of rates of skin and soft tissue infections, pneumonia, and urinary tract infections in older people with diabetes [5]. A recent cohort study of 33,829 people with type 1 diabetes found an association between high mean HbA1c level and their variability with an increased risk of infections, particularly in those who required hospitalization; younger age and non-white ethnicity were associated with a higher infection risk, likely due to higher mean HbA1c level and variations observed in these subgroups [6]. Moreover, a systematic review and meta-analysis suggested an association between higher HbA1C levels and increased risk of surgical site infections (SSIs) after total joint arthroplasty, highlighting that optimizing glycemic control preoperatively and during the postoperative period is crucial for reducing the incidence of SSIs and improving overall surgical outcomes [7]. However, some studies have also reported the impact of low HbA1c levels on infection risk. A population-based cohort study found an increased risk of infection at both high and low HbA1c levels in patients with T2DM [8]. A large primary care cohort study reported that compared with people without diabetes, those with diabetes and poor glycemic control (HbA1c ≥ 11%), but also those with optimal glycemic control (mean HbA1c 6–7%), exhibited elevated hospitalization risks due to infection [9]. Similarly, a prospective cohort study of Chinese adults with T2DM found a U-shaped association between glycaemia and hospitalization for infection, such that HbA1c below 6.0% (42 mmol/mol) and above 8.0% (64 mmol/mol) were both associated with an excess of hospitalization for IDs [10].

Due to the increased risk of infections in diabetic patients, it is important to consider in particular the burden of Vaccine Preventable diseases (VPDs), such as influenza, pneumococcal disease, tetanus, pertussis, herpes zoster, Human papillomavirus (HPV) infection, and COVID-19, that could be prevented in terms of incidence rates and severity through vaccination. People affected by diabetes are at higher risk of developing serious complications from several VPDs when compared to the general population. Previous studies registered an increased mortality rate due to pneumococcal infections among diabetic patients, but whether this is due to a weakened host defense mechanisms against pneumococcus, metabolic decompensation brought on by infection, or other unknown aspects of their multisystem disease is still not clear [11]. Although studies conducted in the past years among patients with DM found impaired or weak antibody response to pneumococcal antigens [12], more recent studies reported normal concentrations of serum antibodies with comparable responses to pneumococcal vaccination between people affected by diabetes and nondiabetic controls [11], as well as the absence of a disparity in the immune response to intramuscular hepatitis B vaccination between children with T1DM and control groups [13]. Conversely, research has shown that compared to people without diabetes, patients with diabetes, particularly those with uncontrolled blood glucose levels, typically have a much weaker antibody response to COVID-19 vaccinations. According to a study by Boroumand et al., patients with T2DM had significantly lower positive rates and titers of specific antibodies (anti-N/S IgG and anti-RBD IgG) than controls. Compared to diabetic individuals with uncontrolled glycemia, those with regulated glycemia showed improved antibody responses [14]. Interestingly, the antibody response to influenza vaccines in diabetic individuals does not appear to be significantly impaired, indicating a different immune response depending on the type of vaccine and the underlying mechanisms of the disease being vaccinated against [15].

The tendency to be more prone to infection, indeed, has a myriad of causes other than defects in immunity. In individuals affected by diabetes, an increased adherence of microorganisms to diabetic cells, the presence of micro- and macroangiopathy or neuropathy, and the elevated frequency of medical interventions may contribute to the pathogenesis of infections, in addition to the tendency toward ketoacidosis, which in itself is complicated by an infection in 75% of cases [16]. A recent meta-analysis of 38 observational studies reported that people with diabetes exhibited significantly higher rates of mortality and hospitalization for influenza compared to the general population. However, lack of data on glycemic control did not make it possible to identify people at higher risk of adverse outcomes among those with diabetes [17]. Moreover, people with diabetes are at higher risk of developing invasive pneumococcal disease, community-acquired pneumonia (CAP) [18,19], as well as of hospitalization due to pneumonia, a longer hospital stay, and higher mortality [20]. Compared with nondiabetic people, those with diabetes and a HbA1 level <7% and those with a HbA1 level ≥9% both had a higher risk of pneumonia-related hospitalization [21]. A multicentric study reported an association between an increased HbA1c level at admission and the development of severe CAP in patients with diabetes [22]. In contrast, a retrospective study on diabetic patients admitted to hospital with CAP found that only those with good glycemic control had an increased risk for adverse outcomes and longer intensive care unit and hospital stays [23]. Diabetes represents an established risk factor for developing severe COVID-19, as well as for hospitalization and mortality from COVID-19 [24]. People with diabetes also are also at increased risk for Hepatitis B (HBV) infection compared to those without diabetes due to their frequent percutaneous exposure to infected blood [25]. The co-presence of T2DM and chronic HBV infection is associated with a higher incidence of hepatocellular carcinoma (HCC), as well as an increased risk of cirrhosis, liver fibrosis, decompensation, non-liver cancers, and all-cause mortality [26]. As such, HZ incidence data and population figures estimate up to 1 million new cases of HZ each year in the United States, with more than half occurring in individuals aged 60 and older [27]. Most countries anticipate an increase in HZ incidence in the ensuing decades as a result of aging populations and the emergence of immunocompromising illnesses (and their treatments) linked to aging [27]. One of the most common complications of HZ is the postherpetic neuralgia (PHN), defined as any pain after the rash heals or any pain at 1 month, 3 months, 4 months, or 6 months after the rash onset. Most experts agree that pain persisting 90–120 days after the onset of the rash defines a group of patients with true chronic neuropathic pain [28]. The prevalence of PHN in the United States is estimated to range from 500,000 to 1 million cases, and people with diabetes are at a higher risk of developing HZ as well as PHN [29]. Despite some studies having highlighted the impact of glycosylated hemoglobin levels on some VPDs, no systematic review has yet analyzed and synthesized these findings.

Therefore, this systematic review aims to evaluate the association between glycosylated hemoglobin levels and VPDs in the adult population.

## 2. Materials and Methods

This systematic review and meta-analysis was conducted in accordance with the updated 2020 Preferred Reporting Items for Systematic Reviews and Meta-Analyses (PRISMA) guidelines [30]. The protocol has been registered in Open Science Framework (https://osf.io/8kxcv/) on 1 November 2023.

### 2.1. Search Strategy

Two independent reviewers (F.L. and E.D.V.) searched PubMed and Embase from inception until 16 November 2023. The complete search strategy is reported in Appendix A. For these datasets, we used the concept of VPDs for HbA1c and the outcomes of interest. Any discrepancies in the literature search were resolved by a third independent investigator (N.V.).

### 2.2. Inclusion and Exclusion Criteria

The PICOS question with which we based our inclusion and exclusion criteria was:P: Adult population (≥18 years) with or without vaccine-preventable diseases (VPDs) at the baseline;I: Higher HbA1c values;C: lower HbA1c values;O: Mortality and hospitalization rates in patients with VPDs and incidence of VPDs in people without VPDs at the baseline;S: Both prospective and retrospective observational studies.

### 2.3. Data Extraction and Risk of Bias

Two authors (F.L. and E.D.V.) independently extracted the following data: name of the first author, date of publication, country of origin, study design, participant age, sample size, gender, VPD considered, follow-up (reported in months), outcome, and confounders used in the multivariate analyses. Any disagreements between reviewers were resolved by a third independent reviewer (N.V.).

The risk of bias was assessed using the Newcastle–Ottawa Scale (NOS) [31]. NOS assigns a maximum of 9 points based on three quality parameters: selection, comparability, and outcome. This evaluation was carried out by two investigators (F.L. and E.D.V.). The risk of bias was classified as high (<5 points), moderate (6–7 points), or low (8–9 points) [32].

### 2.4. Statistical Analysis

Due to the limited number of studies included, we reported the data only descriptively.

## 3. Results

### 3.1. Literature Search

The preliminary literature search yielded 823 non-duplicate publications. After excluding duplicates and abstracts, three full texts were deemed eligible for inclusion in the systematic review. The complete PRISMA flow diagram is shown in Figure 1. At full-text level, the main reasons of exclusion were the lack of the outcome of interest, as well as the lack of population and study design, as detailed in Appendix A.

### 3.2. Descriptive Data

Table 1 shows the descriptive characteristics of the three studies included [33,34]. Overall, the three studies included a total of 705,349 participants without VPDs at the baseline, followed-up for a mean of 113 months. The mean age was less than 65 years in all of the studies included, and the participants were prevalently females (54%). Two studies explored the incidence of HZ infection as outcome, whilst the last one explored the mortality rate according to different HbA1c levels in patients with COVID-19 at the baseline. In the multivariable analyses, as reported in Table 1, the mean number of covariates used was eight, usually considering demographic factors, comorbidities, and medications in exploring the association between basal HbA1c levels and the outcomes of interest.

### 3.3. Main Findings

Table 2 summarizes the main findings of the three studies included in this systematic review. In the study of Bo-Lin Pan et al. [35], in more than 120,000 Chinese participants followed-up for six years, the authors found that higher HbA1c levels at the baseline (>10.3%) were associated with a significantly higher risk of HZ of 44% compared to those with a good HbA1c control (6.7%).

On the contrary, in the study of Kobayashi et al. [33], when compared to the reference group (HbA1c of 5.0–6.4%), participants with a HbA1c less than 5.0% were at significantly higher risk of HZ of 63% after considering the role of 13 potential confounders, whilst participants with a HBA1c more than 9.5% had a similar risk.

Finally, in the UK Biobank dataset, De Jong et al. [34] observed that diabetes, defined using a value of HbA1c more than 7.5%, was associated with a higher risk of death in COVID-19 or influenza/pneumonia only in men, suggesting a potential gender difference.

### 3.4. Risk of Bias

The risk of bias is fully reported in Appendix A. The overall quality of the studies was fair. In all of the three studies, the representativeness of the exposed cohort was poorly described, as was the description about participants lost at follow-up.

## 4. Discussion

In our systematic review examining the relationship between glycosylated hemoglobin levels and VPDs, we identified three full texts eligible for inclusion, encompassing a total of 705,349 participants without VPDs at baseline followed-up for a mean of 113 months. Two studies evaluated the incidence of herpes zoster and one the mortality rate according to different HbA1c levels in patients with COVID-19 at the baseline. Of all of the participants included in our review, 54% were females, with the mean age across the three studies being below 65 years.

To the best of our knowledge, this review is the first to explore the relationship between diabetes and VPDs in terms of mortality and hospitalization rates. The two studies under review evaluating the incidence of HZ were conducted in Asia, specifically in China and Japan. The incidence of T2DM in Japan presents with high heterogeneity and uncertain data, as highlighted by a systematic review and meta-analysis by Goto et al. [36]. Conversely, China ranks among the top three countries worldwide for the highest estimated number of incident diabetes cases, followed by India and the United States of America [37].

The study with the oldest participant cohort, conducted in China by Bo-Lin Pan, reported an increased risk of HZ in patients with poor diabetes control, after adjusting for age, sex, and comorbidities [35]. A systematic review and metanalysis conducted in Taiwan in 2021 found a similar result; nevertheless, in addition to the higher incidence of shingles among diabetic patients compared to the general population, the study found that the substantial difference in risk of occurrence of HZ between diabetics and general population was seen especially in younger patients. In particular, when the cutoff was lowered to 40 years old, there still was an increased risk of diabetic patients compared to the group of non-diabetic peers. However, a limitation of this review was the inclusion of studies that exclusively enrolled patients with TDM1, narrowing the generalizability of the findings and limiting the applicability of the conclusions to the broader diabetic population, particularly those with type 2 diabetes mellitus [38]. A similar result was previously experienced in research conducted in Israel by Heymann et al. [39], in which the risk of HZ in people with a high level (>8% or >64 mmol/mol) of glycated hemoglobin was higher in the population under 45 years old, while such a trend was not found in people of older ages; however, this could be explained by the evidence that younger patients included in that study had more immunodeficiency factors and were less vaccinated. Similarly, a study by Kobayashi et al. in Japan found a significantly higher adjusted odds ratio for developing HZ in the lowest HbA1c group compared to the reference group, while the group with HbA1c ≥ 9.5% showed a higher but not statistically significant odds ratio [33]. The age difference between groups (mean age 39.7 vs. 52.2) and the Japanese vaccination recommendations for individuals over 50 could account for these findings.

Studies provide conflicting data on whether the age-adjusted incidence of HZ in the general population is changing over time [27]. In studies published before the year 2000, the incidence of HZ in people of all ages was 1.2–4.8 cases per 1000 inhabitants per year, and the incidence of HZ in persons over 60 years of age was 7.2–11.8 cases per 1000 inhabitants per year [40]. Numerous research indicates a significant increase in the risk of HZ beginning at 50–60 years of age and continuing to grow into later life [41].

The impact of HZ in diabetic populations has been widely studied globally, although the specific roles of hypo- or hyperglycaemia remain under debate [42,43]. In fact, while the role of hyperglycaemia in the occurrence of infectious diseases has been widely demonstrated, a determining role as a risk factor for infectious events such as HZ is also played by hypoglycaemia, exacerbating immune dysfunctions in diabetic patients inducing stress responses and inflammatory patterns [44]. Studies on post-herpetic neuralgia, which is one of the most common complications of the HZ, have been largely conducted, and a relationship with poor glycaemic control has been hypothesized; however, up to this point, direct research focused on this topic is limited. As such, a large study in the USA involving 4.5 million diabetic patients found an 18% adjusted risk of post-zoster pain, underscoring the strong link between diabetes and persistent post-zoster pain [45]. Conversely, in a cross-sectional study conducted in Canada, the 1-year prevalence was higher in patients aged 65 and older (0.67%), and females had a higher 1-year prevalence compared to males (0.35% vs. 0.28%, respectively); however, this analysis did not identify any statistically significant associations between HbA1c levels and subsequent HZ infections in patients with diabetes. The limitations of the study were associated with wide confidence intervals due to the small number of patients with HZ, reducing the statistical power to detect any significant differences. Furthermore, in this study, there was no differentiation between T1DM and T2DM; this increased incidence by age group and female sex has been observed in other studies [27,46]. A retrospective study conducted by Chen P found that when diabetic patients with CAP have an elevated glycemic gap, they are more likely to experience poor outcomes such as acute renal damage (AKI), septic shock, ICU admission, prolonged hospital and ICU stays, and more days spent on a ventilator. The glycemic gap between admission glucose levels and A1c-derived average glucose levels can predict these unfavorable outcomes, performing better than acute hyperglycemia. Clinical scoring systems including PSI, CURB-65, and SMART-COP showed higher predictive accuracy when the glycemic gap was added. This suggests that the glycemic gap could be a useful addition to future clinical scoring systems for enhanced risk assessment and management [23].

Furthermore, the importance of maintaining constant blood sugar levels for a successful immune response has been extensively shown by the fact that diabetic individuals with regulated glycemia showed considerably better antibody responses than those with uncontrolled glycemia. The aforementioned discovery highlights the crucial function of glucose control in enhancing the body’s ability to fight infections and respond to vaccinations, implying that well-managed diabetes can lead to improved health outcomes [14]. The recent literature has increasingly focused on exploring a potential correlation between viral infections and diabetes. However, despite this heightened attention, epidemiological data continue to present significant inconsistencies and challenges [47].

A study by Lutz P. Breitling et al. reported a non-linear association between HbA1c levels and mortality from influenza, pneumonia, or other acute lower respiratory infections, but it was excluded from our meta-analysis due to its lack of differentiation in the results analysis between influenza and non-viral pneumonia patterns [48]. In terms of VPDs, a higher risk of HBV infection in diabetic populations has been well documented, with more severe outcomes than in the general population [49] and a higher incidence of diabetes in individuals infected with HBV [50]. A recent study by T. Cai et al. found that diabetic patients with HBV had significantly worse glycemic control than those with T2DM alone; HBV infection was indeed linked to a 33% increased risk of developing diabetes, making it a potential diabetes-related risk factor and suggesting that HBV treatment could mitigate this effect. However, the study by Cai initially selected for our meta-analysis was later excluded due to the lack of relevant outcomes [48]. Moreover, elevated HbA1c levels were positively related to the risk of HCC in diabetic patients [51]. However, some studies indicates that this increased risk is more related to liver cirrhosis resulting from chronic hepatitis B rather than the HBV infection itself [52]. However, the relationship between T2DM and HBV is still controversial, due to a potential protective effect of HBV infection against T2DM [53]. Among viral VPD an important role is played by HPV; in this regard, many studies have focused on understanding a potential correlation between HPV and diabetes, founding a higher prevalence of HPV infection in people with diabetes compared to the general population [54] and proving the association between diabetes and increased risk of developing cervical cancer, even if a direct correlation with HPV was not proved [55]. None of the studies on HPV were included in our meta-analysis, as they did not meet the inclusion criteria.

The last included study found that diabetes, defined using a value of HbA1c more than 7.5%, was associated with a higher risk of death in COVID-19 only in men, suggesting a potential gender difference [34].

Large-scale studies from China and Italy have shown that diabetic patients have a similar risk of SARS-CoV-2 infection as those with optimal glycaemic control [56,57]. Yet, once hospitalized, diabetic patients exhibit higher morbidity and mortality rates, indicating that diabetes is an independent predictor of poor prognosis in COVID-19 patients. As stated by Tabrizi et al. in a recent study, diabetic individuals infected with SARS-CoV-2 had over twice the risk of ICU admission and more than three times the risk of death compared to non-diabetic patients [58]. Conversely, a study in England by Barron et al. on 61 million individuals found that 33% of in-hospital COVID-19 deaths between 1 March 2020 and 11 May 2020 occurred in people with diabetes, with diabetic women having higher odds ratios for in-hospital mortality than men [59]. A recent systematic review and meta-analysis of 22 studies reported an association between lower HbA1c prior to hospitalization and lower in-hospital mortality in patients with COVID-19 [60]. Thus, the increased vulnerability of people affected by diabetes compared to the general population can lead to more severe illness, longer recovery times, and an increased likelihood of hospitalization. Consequently, it is crucial for individuals with diabetes to maintain up-to-date vaccinations to protect themselves against preventable diseases such as influenza, pneumonia, and hepatitis. Public health initiatives and healthcare providers should emphasize the importance of immunizations in this high-risk group to reduce morbidity and mortality rates.

However, studies on the relationship between diabetes and VPDs are still grappling with diverse findings and interpretations, which highlight the complexity of establishing clear associations between infections and specific health conditions like diabetes. The evolving landscape of research underscores the need for further investigation and comprehensive analysis to elucidate the nuanced relationships between infections and health outcomes across different populations and contexts.

To the best of our knowledge, this review represents a pioneering effort in investigating the relationship between HbA1c levels and VPDs, examining both their incidence in individuals without VPDs and the impact on mortality and hospitalization rates among patients with VPDs. While this study marks significant progress in understanding these correlations, several limitations must be acknowledged and addressed.

The primary limitation of this study is the reliance on only two databases, Embase and PubMed, for sourcing the data. This constraint may still impact the comprehensiveness and generalizability although these databases are among the most comprehensive and widely used in the biomedical field. Furthermore, the review includes only three cohort studies, which may not fully capture the breadth and diversity of potential associations between HbA1c levels and various VPDs. The limited number of studies also restricts the generalizability of findings across different populations and settings. Moreover, case–control studies may have provided limited insights into causal relationships between high HbA1c levels and the outcomes of our interest.

The focus of the review is predominantly on HZ and COVID-19, with a notable exclusion of other VPDs. While HZ and COVID-19 are significant illnesses, broader inclusion of other VPDs could have enriched the review’s scope and implications for public health policies and clinical practices.

Regarding study quality, the included studies generally exhibited low risk of bias. However, concerns were raised about population representativeness, inadequate follow-up rates, and the retrospective nature of some studies. These methodological limitations may introduce biases and affect the reliability and applicability of the findings. Notably, only one of the studies reviewed was prospective, emphasizing the need for more longitudinal research to establish robust conclusions about the impact of HbA1c levels on VPD outcomes over time.

Moving forward, there is a clear imperative for further research in this field to address these limitations and expand our understanding. Future studies should aim for greater diversity in study populations to enhance the generalizability of findings across different demographic and geographic contexts. Longitudinal studies with adequate follow-up periods are crucial to capture the long-term effects of HbA1c levels on VPD incidence, mortality, and hospitalization rates. Additionally, incorporating case–control designs would enable better control of confounding variables and strengthen causal inference.

Furthermore, expanding the scope of research to include a wider range of vaccine-preventable diseases beyond HZ and COVID-19 would provide a more comprehensive picture of how HbA1c levels influence susceptibility and outcomes across different infectious diseases.

## 5. Conclusions

Our systematic review, while limited to a small number of studies, demonstrates a significant association between high HbA1c levels and increased risk of death from COVID-19 in men, suggesting a potential gender difference. For HZ, discrepancies exist due to the significant impact of both hyperglycemia and hypoglycemia on the development of infectious diseases in diabetic populations. As such, emerging evidence indicates that hypoglycemia also plays a crucial role by inducing stress responses and inflammatory patterns that exacerbate immune dysfunctions. Therefore, maintaining stable blood glucose levels is essential to reduce the susceptibility to infections and improve overall health outcomes in diabetic patients, and further prospective studies are needed to explore this correlation more thoroughly. Diabetic individuals are more vulnerable to severe illness, longer recovery times, and higher hospitalization rates compared to the general population. Hence, it is crucial for these individuals to maintain up-to-date vaccinations against preventable diseases like influenza, pneumonia, and hepatitis. By addressing these challenges and expanding the scope and rigor of studies, we can better inform preventive strategies, clinical management, and public health interventions aimed at reducing the burden of VPDs in diabetic populations and beyond. Moreover, the aging population and high prevalence of diabetes in older adults, more vulnerable to infectious diseases and their complications, highlight the need for more research into this high-risk age group.

## Figures and Tables

**Figure 1 diseases-12-00187-f001:**
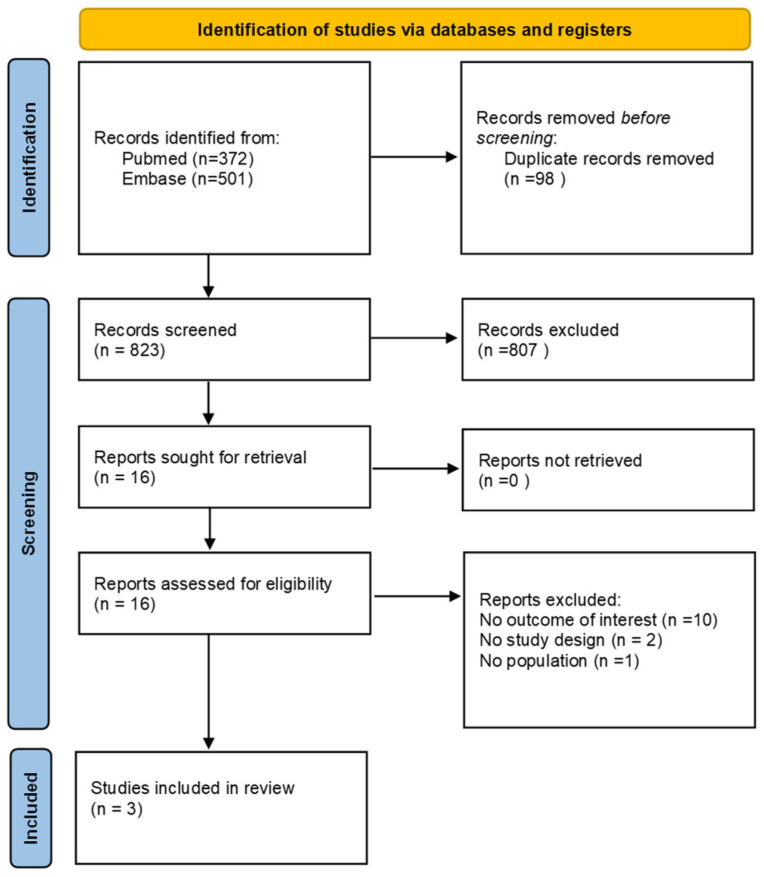
PRISMA flow diagram showing the process of study selection.

**Table 1 diseases-12-00187-t001:** Descriptive characteristics of the studies included.

Author	Year	Country	Type of Study	VPD Considered	Follow-Up (Months)	Total Sample Size	% of Females	Mean Age	Outcome	Confounders Mentioned in the Multivariate Analysis (Number)
Bo-Lin Pan [35]	2022	China	Prospective cohort study	HZ	72	121,999	46.3	60.4	Incidence of HZ	Age, sex, comorbidities (3)
Kobayashi D [33]	2019	Japan	Retrospective longitudinal study	HZ	132	81,466	51.3	46.5	Incidence of HZ	Age, sex, time variable; social histories, including smoking status, alcohol consumption, BMI, and exercise habits; history of diabetes and its pharmacological treatment status, medical histories, including cancer, autoimmune disease, hemodialysis, and chronic hepatitis/cirrhosis (13)
De Jong M [34]	2021	UK	Prospective cohort study	COVID-19	134	501,884	54	Aged 40–69 years at baseline	Death from COVID-19	Age, BMI, socioeconomic status, smoking, systolic blood pressure, total cholesterol, antihypertensivemedication, glucose-loweringmedication, lipid lowering medication (9)
TOTAL			Two prospective; one retrospective		113	705,349	54		Two: HZ; one: COVID-19	8

**Table 2 diseases-12-00187-t002:** Main findings of the studies included.

Author	Year	Main Findings
Bo-Lin Pan et al. [35]	2022	Four HbA1C trajectories were identified: ‘good control’ (mean HbA1C of 6.7% or 50 mmol/mol), ‘high decreasing’ (mean HbA1C of 7.9% or 63 mmol/mol), ‘moderate control’ (mean HbA1C of 8.4% or 68 mmol/mol), and ‘poor control’ (mean HbA1C of 10.7% or 93 mmol/mol). A significantly higher risk of HZ was observed in the ‘poor control’ trajectory with an HR = 1.44 (95% CI 1.26–1.64) after adjusting for confounders and comorbidities.
Kobayashi D [33]	2019	Individuals in the lowest HbA1c group (HbA1c of <5.0%) exhibited a significantly higher risk of developing HZ (OR 1.63; 95%CI: 1.07–2.48) compared with the reference group (HbA1c of 5.0–6.4%). Individuals in the highest HbA1c group (HbA1cof ≥ 9.5%) had a higher but nonsignificant risk than the reference group (OR 2.15; 95% CI, 0.67–6.94).
De Jong M [34]	2021	There was no association between higher levels of HbA1c and increased risk of COVID-19 or influenza/pneumonia death in women. On the contrary, compared with no diabetes, an HbA1c > 7.5% (58 mmol/mol) was associated with an increased risk of COVID-19 or influenza/pneumonia death in men.

## Data Availability

Data are available upon request to the corresponding author.

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
