# Peer review of "Association between Glycosylated Hemoglobin Levels and Vaccine Preventable Diseases: A Systematic Review"

_diseases, 2024, doi:10.3390/diseases12080187_

Round 1
Reviewer 1 Report
Comments and Suggestions for Authors
Thanks for the opportunity to review the manuscript entitled ”Association between Glycosylated Haemoglobin Levels and Vaccine Preventable Diseases: a Systematic Review”.
The authors start from the premise that people with diabetes have higher risks of vaccine-preventable diseases. The articles included in this meta-analysis conclude that both high and low HbA1c levels increase herpes zoster risk, while high HbA1c levels elevate COVID-19 mortality risk in men, meaning that poor control of diabetes is associated with high risk of infections, an information which is not a new finding.
For me, it is not very clear which were exactly the inclusion criteria (should be more details) of the studies included in metaanalysis and the same for exclusion criteria, thus from 823 studies remained only 3 studies (page 4, figure 1?, it is not written the number and name of figure).
I did not see the supplementary tables.
The conclusions are not objective, containing general informations related to the need for a better control of diabetes.
Comments on the Quality of English Language-
Author Response
For me, it is not very clear which were exactly the inclusion criteria (should be more details) of the studies included in metaanalysis and the same for exclusion criteria, thus from 823 studies remained only 3 studies (page 4, figure 1?, it is not written the number and name of figure).
R: Thank you for the comment. We have now better detailed the inclusion criteria.
I did not see the supplementary tables.
R: Sorry for the inconvenience. In the revised version, we have uploaded the supplementary material.
The conclusions are not objective, containing general informations related to the need for a better control of diabetes.
R: Good point. We have now updated the conclusions of our work, as suggested.
Reviewer 2 Report
Comments and Suggestions for Authors
-
The manuscript's literature search was restricted to only two databases, which may limit the comprehensiveness and representativeness of the findings. It is recommended to expand the scope of the literature search by including additional databases to ensure a more thorough review of the available evidence. Given the significant mention of China's high incidence of diabetes globally, it is essential to include searches in Chinese literature databases to capture relevant studies from this region.
-
The manuscript includes only three studies, all of which report differing results. It is crucial to address how the robustness of these findings is ensured. The conclusion drawn from such a limited dataset raises concerns about the reliability of the results.
-
While the supplementary materials provide details on the literature retrieval process, the manuscript should clearly outline the search strategy including specific keywords used. This transparency is essential for readers to understand the methodology employed in retrieving relevant information.
-
The inclusion and exclusion criteria require more detailed elaboration to ensure transparency and comparability across studies. Specifically, the PICOS framework used for defining these criteria should be clarified to enhance understanding of the study methods.
-
There appears to be an issue with the sequencing of references in the manuscript. Reference 61 is listed before references 41, 42, and 43 in the reference list, which needs to be corrected for clarity and consistency.
-
The manuscript prominently features the term "vaccine-preventable diseases (VPDs)" in the title and keywords but lacks a clear definition. It is essential to provide a specific definition of VPDs within the context of this study. Furthermore, clarification is needed on how VPDs such as Herpes Zoster and COVID-19 relate to the study's focus and findings.
Minor editing of English language required
Author Response
The manuscript's literature search was restricted to only two databases, which may limit the comprehensiveness and representativeness of the findings. It is recommended to expand the scope of the literature search by including additional databases to ensure a more thorough review of the available evidence. Given the significant mention of China's high incidence of diabetes globally, it is essential to include searches in Chinese literature databases to capture relevant studies from this region. The manuscript includes only three studies, all of which report differing results. It is crucial to address how the robustness of these findings is ensured. The conclusion drawn from such a limited dataset raises concerns about the reliability of the results.
R: We fully agree with this consideration. Unfortunately, no one of the authors know Chinese. For example, we have tried to screen title/abstracts in the Chinese Biomedical Databases, but many of them are not reported in Western characters. Since it is a major limitation, we have recognized this in the Limitations section:
“The primary limitation of this study is the reliance on only two databases, Embase and PubMed, for sourcing the data. This constraint may still impact the comprehensiveness and generalizability although these databases are among the most comprehensive and widely used in the biomedical field.”
While the supplementary materials provide details on the literature retrieval process, the manuscript should clearly outline the search strategy including specific keywords used. This transparency is essential for readers to understand the methodology employed in retrieving relevant information.
R: Sorry for the inconvenience. We have now reported some more details about the main keywords used.
The inclusion and exclusion criteria require more detailed elaboration to ensure transparency and comparability across studies. Specifically, the PICOS framework used for defining these criteria should be clarified to enhance understanding of the study methods.
R: Thank you for the comment. We have now better detailed the inclusion criteria.
There appears to be an issue with the sequencing of references in the manuscript. Reference 61 is listed before references 41, 42, and 43 in the reference list, which needs to be corrected for clarity and consistency.
R: Sorry for the inconvenience. The reference list is now in the right order.
The manuscript prominently features the term "vaccine-preventable diseases (VPDs)" in the title and keywords but lacks a clear definition. It is essential to provide a specific definition of VPDs within the context of this study. Furthermore, clarification is needed on how VPDs such as Herpes Zoster and COVID-19 relate to the study's focus and findings.
R: We have now added in the Introduction section the definition of VPDs, as required:
“Due to the increased risk of infections in diabetic patients, it is important to consider in particular the burden of Vaccine Preventable diseases (VPDs), such as influenza, pneumococcal disease, tetanus, pertussis, herpes zoster, Human papillomavirus (HPV) infection, COVID-19, that could be prevented in terms of incidence rates and severity through vaccination.”
Round 2
Reviewer 1 Report
Comments and Suggestions for Authors
Thanks for the opportunity to review the revised version of the manuscript entitled ”Association between Glycosylated Haemoglobin Levels and Vaccine Preventable Diseases: a Systematic Review”.
The authors made efforts to add the data and informations requested in the first revision.
I further suggest only some corrections of minor errors or text editing.
Comments on the Quality of English Language-
Author Response
Thank you so much :)
Reviewer 2 Report
Comments and Suggestions for Authors
My primary concerns have still not been adequately addressed. Therefore, I cannot accept it.
Comments on the Quality of English Languageneed to improve
Author Response
Round 2: My primary concerns have still not been adequately addressed. Therefore, I cannot accept it.
R: We are very sorry that we are not able to completely answer to the pertinent questions raised by the Reviewer 2. However, it seems to us that we are unable to answer to only one question, i.e., the lack of databases from China. As explained in our revision and added in the manuscript, unfortunately, no one of the Authors know Chinese and it is essential to access these databases. At the same time, as mentioned, in the answer we believe that no seminal papers are missing in our systematic review that included the screening of thousands of papers.